# Inflammation of Dry Eye Syndrome: A Cellular Study of the Epithelial and Macrophagic Involvement of NFAT5 and RAGE

**DOI:** 10.3390/ijms241311052

**Published:** 2023-07-04

**Authors:** Fanny Henrioux, Valentin Navel, Corinne Belville, Coline Charnay, Audrey Antoine, Frédéric Chiambaretta, Vincent Sapin, Loïc Blanchon

**Affiliations:** 1Team “Translational Approach to Epithelial Injury and Repair”, Institute Genetics, Reproduction and Development (iGReD), Université Clermont Auvergne, 63000 Clermont-Ferrand, France; fanny.henrioux@uca.fr (F.H.); vnavel@chu-clermontferrand.fr (V.N.); corinne.belville@uca.fr (C.B.); coline.charnay@uca.fr (C.C.); audrey.antoine@uca.fr (A.A.); fchiambaretta@chu-clermontferrand.fr (F.C.); vincent.sapin@uca.fr (V.S.); 2Ophthalmology Department, CHU Clermont-Ferrand, 63000 Clermont-Ferrand, France; 3Biochemistry and Molecular Genetics Department, CHU Clermont-Ferrand, 63000 Clermont-Ferrand, France

**Keywords:** dry eye, RAGE, NFAT5, inflammation, macrophages

## Abstract

Dry eye inflammation is a key step in a vicious circle and needs to be better understood in order to break it. The goals of this work were to, first, characterize alarmins and cytokines released by ocular surface cells in the hyperosmolar context and, second, study the role of NFAT5 in this process. Finally, we studied the potential action of these alarmins in ocular surface epithelial cells and macrophages via RAGE pathways. HCE and WKD cell lines were cultured in a NaCl-hyperosmolar medium and the expression of alarmins (S100A4, S100A8, S100A9, and HMGB1), cytokines (IL6, IL8, TNFα, and MCP1), and NFAT5 were assessed using RT-qPCR, ELISA and multiplex, Western blot, immunofluorescence, and luciferase assays. In selected experiments, an inhibitor of RAGE (RAP) or NFAT5 siRNAs were added before the hyperosmolar stimulations. HCE and WKD cells or macrophages were treated with recombinant proteins of alarmins (with or without RAP) and analyzed for cytokine expression and chemotaxis, respectively. Hyperosmolarity induced epithelial cell inflammation depending on cell type. NFAT5, but not RAGE or alarmins, participated in triggering epithelial inflammation. Furthermore, the release of alarmins induced macrophage migration through RAGE. These in vitro results suggest that NFAT5 and RAGE have a role in dry eye inflammation.

## 1. Introduction

Dry eye disease (DED) is a common ocular surface disorder that is defined as “a multifactorial disease of ocular surface characterized by a loss of homeostasis of the tear film (TF), and accompanied by ocular symptoms, in which tear film instability and hyperosmolarity, ocular surface inflammation and damage, and neurosensory abnormalities play etiological roles” by the TFOS (DEWS II). [1]. DED affects millions of people worldwide, with different degrees of severity ranging from discomfort to pain. It has an important impact on vision-related quality of life by limiting certain activities, such as night driving [2].

The onset of DED is strongly associated with tear hyperosmolarity and subsequent sterile inflammation. According to the well-known concept of the “vicious cycle of inflammation” [3], TF instability and hyperosmolarity induce excessive stress on the underlying epithelia (cornea and conjunctiva), which leads to the release of numerous molecules such as alarmins (damage-associated molecular pattern/DAMPs), cytokines, and metalloproteinases. This phenomenon leads to TF instability such as hyperosmolarity and contributes to feeding the previously described “vicious cycle.”

More specifically, the inflammatory signaling molecules alarmins (DAMPs) released during a sterile inflammation are recognized by pattern recognition receptors (PRRs). DAMPs include high-mobility group box 1 (HMGB1) protein, the S100 proteins family, uric acid, cell-free DNA, and advanced glycation end-products (AGEs). The most studied PRRs include toll-like receptors (TLRs), scavenger receptors, NOD-like receptors, and the receptor for AGEs (RAGE) [4]. Following such recognition, PRRs lead to a microbe-free inflammatory response or a “sterile” inflammation. Thus, it has been determined that TF, in the case of DED, contains many of these alarmins, such as S100A4, S100A6, S100A8, S100A9, S100A11, and HMGB1 and many cytokines, which are accompanied by a recruitment of numerous immune cells in the ocular surface [5,6,7,8]. Moreover, regarding PRRs, they seem to have an increase of TLR expression and recruitment, such as TLR4 and TLR2, in surface epitheliums [9,10,11,12], with a particular emphasis on TLR4 implication in inflammation and on immune cell recruitment [11]. However, these obtained results do not clearly demonstrate the link between the alarmins and PRRs observed in this inflammation. It also remains unclear how this sterile inflammation is triggered and can pass from an acute state to a chronic one. Owing to the fact that sterile inflammation is a core driver of DED, these knowledge gaps need to be closed in order to investigate how to break this vicious cycle as probably one of the most important steps in the medical management of DED [8].

In this study, we focused our work on two molecular actors of sterile inflammation: NFAT5 and RAGE. NFAT5 is a transcription factor known to act against hyperosmolarity through the activation of osmoprotectants and inflammatory molecules. To achieve this, NFAT5 is quickly upregulated in corneal and conjunctival epithelium in the hyperosmolarity context and seems to be implicated in cytokine expression [13,14]. In addition to this, RAGE, a cell surface receptor which interacts with several ligands of the alarmin type (including S100 protein family and HMGB1), is known to be implicated in the pathogenesis of many inflammatory diseases through the activation of pro-inflammatory pathways [15,16,17]. This activation will induce cytokine expression and release and the attraction of immune cells to an inflammation site [18,19,20]. In dry eye syndrome, many potential RAGE ligands are released [5,6,7], but their involvement has not yet been studied at the ocular epithelium level. Thus, our study intends to provide new information about the molecular actors implicated in inflammation by focusing on the similarity/difference of response of surface epithelial cells (cornea and conjunctiva) in triggering the DED inflammation. To achieve this, we investigate in detail the implication of NFAT5 in the acute inflammation of the epithelial cell surface and the activation of the RAGE–ligands axis in the development of chronic inflammation through migration and recruitment of macrophage immune cells.

## 2. Results

### 2.1. Effects of Hyperosmolarity on Ocular Surface Epithelial Cells

The mRNA and protein expression profiles of proinflammatory cytokines (MCP1, IL6, IL8, and TNFα) in surface epithelial cells (human corneal epithelial [HCE] and Wong–Kilbourne derivative [WKD]) were investigated. RT-qPCR experiments revealed that HCE and WKD expressed differential cytokines after 24 h under hyperosmolarity stress. For WKD cells, an increase in IL6, IL8, TNFα, and MCP1 mRNA expressions was revealed, with the last one particularly pronounced in the 90 mM NaCl condition (Figure 1A). By contrast, in HCE cells, we observed a significant increase in TNFα and MCP1 mRNA expression (Figure 1B). These results and the cell specificity answer were confirmed by the release of protein levels (Figure 1C,D) using Multiplex and ELLA technology. Such an increase is particularly important for MCP1 and in both cell lines.

The same type of study was carried out to investigate the implication of surface epithelial cells in the expression and release of DAMPs (S100A8, S00A9, S100A8/S100A9, and S100A4) in a hyperosmotic condition in HCE and WKD cells. RT-PCR experiments again revealed a cell-type-specific response between HCE and WKD cells. For WKD cells, we observed an increase in S100A8 (significant for 90 mM NaCl), S100A9, and S100A4 mRNA expressions (Figure 2A), and for HCE, a sole increase of S100A4 expression was significant at the 90 mM NaCl condition (Figure 2B). These results were confirmed by the release of protein levels (Figure 2C,D). In addition, the presence of S100A8/S100A9 heterodimer (calprotectin) in the medium was studied, as the previous results showed an increase of each of these proteins separately. Nevertheless, the results did not show a statistical increase of this heterodimer for the two cell lines in the hyperosmolar condition (Figure 2C,D). Besides the S100 family member proteins, another DAMP, HMGB1, was studied in HCE and WKD. The release of HMGB1 was statistically increased in both cells at 12 and 24 h and in the 90 mM NaCl condition (Figure 3).

### 2.2. Characterization of NFAT5 in Ocular Surface Epithelial Cells in Hyperosmolar Stress Condition

In addition to the release of pro-inflammatory cytokines and DAMPs, the involvement of NFAT5 in the inflammatory response elicited by hyperosmolarity in HCE and WKD cells was studied. We first investigated the effect of hyperosmolar stress on NFAT5 expression. The RT-qPCR experiments on WKD and HCE revealed a specific time increase of NFAT5 mRNA after 6 h for the 90 mM condition and after 24 h for the 70 mM condition (Figure 4A,B, respectively). This mRNA overexpression was an associated consequence of the increase of NFAT5 protein level in WKD (Figure 4C), but only at 12 h for HCE, which appears to be cell- and time-specific (Figure 4D).

Considering that NFAT5 is a transcription factor, we initially investigated its potential nuclear relocalization and transcriptional activity in both cell lines. First, we clearly observed via immunofluorescence a stronger intensity of NFAT5 (green) localized in the nuclei under hyperosmolar conditions for both cell types after 6 h (Figure 5A,B). This observation was likewise confirmed via luciferase assays, which revealed an increase of NFAT5 regulatory activity at 6, 12, and 24 h in hyperosmolar conditions for HCE and WKD as attested by the increase of luciferase amount (Figure 5C,D, respectively).

### 2.3. Implication of NFAT5 in Surface Epithelial Cell Inflammation

To investigate the direct involvement and importance of NFAT5 in the inflammation status of surface epithelial cells, HCE and WKD cells were transfected with NFAT5 small interfering RNA (siRNA). RNA analysis of NFAT5 showed a decrease in NFAT5 level and thus confirmed the action of the siRNA (Appendix A). After 24 h of NFAT5 siRNA transfection and 24 or 48 h of hyperosmolar medium, the mRNA expression and protein release of pro-inflammatory cytokines (MCP1, IL6, IL8, and TNFα) were analyzed in accordance with each cell type using the results obtained in Figure 1. The inhibition of NFAT5 led to a decrease of MCP1 mRNA and protein release for WKD compared to the same condition without the inhibition of NFAT5 (Figure 6A,C). For all other cytokines (IL6, IL8, and TNFα), the inhibition of NFAT5 had no effect on hyperosmolarity-induced cytokine expression or had a weak, not statistically significant effect. In HCE cells, NFAT5 inhibition led to a decrease of MCP1 and TNFα mRNA (Figure 6B). These mRNA results were confirmed at the protein level for MCP1, whereas for TNFα protein levels, a non-statistically significant decrease was underlined but could be interpreted as a time-dependent tendency (Figure 6D).

Knowing that NFAT5 acts quickly, we also studied its direct involvement in alarmin expression and, more precisely, S100 proteins (mRNA and protein release). The RT-qPCR revealed for WKD a decrease of S100A9 and S100A4 expressions in hyperosmolar stress conditions (principally in 90 mM), with NFAT5 inhibition compared to hyperosmolar conditions without inhibition (Figure 7A). As for TNFα, the protein quantification did not show a significant decrease (but a tendency) of S100A9 with NFAT5 inhibition but with the same trend as in mRNA. Nevertheless, for S100A4, the protein results showed a decrease of S100A4 with the inhibition of NFAT (Figure 7C). Since HCE cells overexpress only S100A4 (Figure 2), we studied NFAT5 implication only for this DAMP expression, which showed a decrease of this S100A4 protein release (principally in 70 mM) with NFAT5 inhibition at the 70 mM NaCl condition but just a trend for mRNA (Figure 7B,D).

### 2.4. RAGE–Ligands Axis on Ocular Surface Epithelial Cell Inflammation

Since we observed DAMPs being released by epithelial cells under hyperosmolar stress, our first hypothesis was to test the potential involvement of RAGE, as a PRR receptor, in the inflammation of these epithelia (“autocrine action”). In addition to the hyperosmolar stress of HCE and WKD, cells were exposed concomitantly to a RAGE inhibitor (RAP) for either 24 or 48 h. Figure 1 indicates that mRNA expression and protein release of pro-inflammatory cytokines (MCP1, IL6, IL8, and TNFα) overexpress the same cytokines under hyperosmolar stress (Figure 8). With the addition of the inhibitor RAP, there were no significant differences in cytokine expression in either cell line (Figure 8). Following such results concerning RAGE, the same type of experiments were repeated to test the potential implication of the classical pro-inflammatory receptors TLR4 using a specific inhibitor for this one. The same results were obtained and detailed in Appendix A.

In addition, we treated epithelial cells with recombinant purified proteins to test whether or not such released alarmins could directly induce surface epithelial cell inflammation via PRRs in an autocrine manner. Thus, the mRNA and protein expression profiles of pro-inflammatory cytokines (MCP1, IL6, IL8, and TNFα) in surface epithelial cells (HCE and WKD) were investigated with alarmins (S100A4, S1008, S100A9, and HMGB1) treated separately or with normal medium/non-treated (NT) cells. We tested two concentrations for each alarmin, and RT-qPCR and protein release data revealed a non-significant increase of the different cytokines studied compared to NT cells: TNFα and MCP1 for HCE and IL6 IL8, TNFα and MCP1 for Chang cells (see Appendix A for mRNA results and Figure 9 for protein results).

### 2.5. Paracrine Effect of DAMPs and MCP1 on Macrophage Migration

We were interested in the potential impact of the discovered DAMPs on chronic inflammation. Indeed, as mentioned above (Section 2.4, Figure 9), the discovered DAMPs do not have an autocrine impact on ocular surface epithelial cells. Moreover, among the cytokines released, the presence of MCP1 could open a hypothesis about its action on macrophages. We therefore studied the potential of these DAMPs and MCP1 in macrophage attraction to the inflammatory site. Using inserts, we measured the number of macrophages crossing the insert in the presence or absence of DAMPs (S100A4, S100A8, S1009, and HMGB1) or MCP1 and with or without RAP (RAGE inhibitor) and TAK (TLR4 inhibitor). Our results revealed that the presence of S1004, S1009, and HMGB1 independently increased macrophage migration. This was also the case for MCP1, the chemokine known to attract monocytes/macrophages, which could be considered a positive control of chemotaxis (Figure 10A). In the presence of the RAP inhibitor, we also observed a significant inhibition of macrophage migration for each DAMP studied and with a chemokine action (S100A4, S100A9, and HMGB1), revealing the influence of RAGE on such a migratory effect. The TAK inhibitor had no inhibitory impact on macrophage migration in the presence of S100A4 and HMGB1, whereas this one inhibited the action of S100A9.

## 3. Discussion

Dry eye syndrome is a multifactorial disease with multiple sub-types of dry eye. It is classically defined as a common ocular surface disease with TF hyperosmolarity and instability and a chronic vicious circle of local inflammation. Breaking this vicious cycle is an important step in the treatment of DED, and sterile inflammation seems to be the core driver of DED. The conjunctiva and cornea are important tissues to study as they cover a major part of the ocular surface where inflammatory reactions take place. Moreover, conjunctival and cornea cells themselves can secrete globally inflammatory cytokines and participate in inflammatory processes [13,14]. In this work, we used an epithelial conjunctival (WKD) and corneal (HCE) cell line in a hyperosmolar NaCl-induced in vitro model of dry eye in order to precisely identify the function of each cell type.

Studying inflammatory profiles in a hyperosmolar context allows us to highlight for the first time that the release of molecular actors involved in the activation of this inflammation (IL6, IL8, TNFα, and MCP1) is cell specific when comparing conjunctival and corneal cells. Using our cellular model, we also confirmed the activation and involvement of NFAT5 in dry eye sterile inflammation. NFAT5 is a fundamental regulator of the response to osmotic stress in mammalian cells [21]. This gene was initially identified in 1999 as a novel member of the Rel family, which comprises NFκB and NFAT proteins [22]. Since its discovery, NFAT5 has been analyzed mostly in the context of the hypertonicity stress response. Furthermore, the advent of mouse models deficient in NFAT5 and other recent advances have confirmed a fundamental osmoprotective role of this factor in mammals [21] and revealed features that suggest a wider range of functions, such as inflammatory cytokine production and immune cell development [23,24]. Thus, our results showed an increase of NFAT5 expression (mRNA and protein), a nuclear localization from cytoplasm (via immunocytochemistry), and an increase of NFAT5 activation in a hyperosmolar context. As hyperosmolar stress induced NFAT5 activation in as little time as 6 h in HCE and WKD cells in our experiments, NFAT5 could be considered an early-response gene in terms of transcription/translation on its own and on the emergence of an acute inflammatory stress. The inhibition of NFAT5 also abolished the overexpression and production of MCP1 (CCL2) in both cell types. BY contrast, only for the corneal cell (HCE), this inhibition seemed to decrease the expression of TNFα. Again, such data reveal that both cell types have their specific action in terms of reacting to hyperosmolar conditions. The involvement of NFAT5 in MCP1 regulation of expression in the conjunctiva has already been demonstrated [13], but our work demonstrates, for the first time, the same regulation in the corneal context.

In addition to the fact that it seems clear now that NFAT5 is indeed involved in ocular surface inflammation, such cellular phenomena also contribute to the production of specific molecules called alarmins (or DAMPs), which are expressed and produced endogenously when the cells suffer. Indeed, hyperosmolarity induces significant stress at the ocular surface, resulting in the production of DAMPs. For example, S100 proteins are normally intracellular proteins involved in various cellular functions, such as calcium homeostasis, cell growth and differentiation, cytoskeleton dynamics, and energy metabolism [25,26,27]. However, when cells are in danger, they release these types of proteins during cell death or by secretion, proteins that have been found in the tears of patients with DED [5,6]. HMGB1, another DAMP that is normally a nuclear protein involved in DNA repair, can become an alarmin in case of cell aggression, triggering an inflammatory cascade [28,29,30]. Again, HMGB1 was also found to be increased in the tears of patients with dry eye syndrome compared to those of healthy patients [31]. These initial data highlighted the presence of alarmins in sterile inflammation, which is probably directly linked to the vicious circle of dry eye syndrome. However, knowledge is still lacking about which receptor recognizes these alarmins, the cellular origin of these alarmins, and their inflammatory impact in this pathology. Thus, our results, using the same model of hyperosmolarity and cell types, revealed an increased expression and release of S100 proteins (S100A4, S100A8, and S100A9), which are, again, cell-dependent (conjunctival or corneal). However, HMGB1 release is not cell-specific due to the ubiquitous expression of this protein in human cells. This last result is not really surprising, considering the fact that this protein is originally in a nuclear state that evolves quickly into a secreted alarmin (without neo-production) in case of cell aggressions. As for the inflammatory molecules, we have determined the impact of NFAT5 on the expression of these alarmins (only those overexpressed in a hyperosmolar context). Thus, NFAT5 inhibition abolished the overexpression and production of S100A4 in both cell types and seems to be the same for S100A9 in WKD cells. Taken together, these first results, according to our aggressive hyperosmolar model, demonstrate that such condition will lead to the production and secretion of inflammatory molecules and alarmins by the global ocular surface epithelium. Furthermore, it demonstrates a cell-specific response between conjunctival and corneal cells in such a cellular response and underlines the importance of the transcription factor NFAT5. For us, following the model used, this first part of the study constituted a clarification and update as to how an acute inflammatory state could be triggered in the case of DED. Nevertheless, understanding the evolution to a chronic state that is so deleterious in the case of dry eye constitutes a major challenge in facilitating at term the medical management of this pathology. This switch from one state to another has to be investigated, and the actions of alarmins previously described (S100A4, S100A8, S100A9, and HMGB1) could be of primary importance.

For this purpose, we then focused our work on one well-known membranous receptor called RAGE. RAGE is a glycoprotein transmembrane receptor binding DAMPs (also called alarmins): the HMGB1/amphoterin family and S100/calgranulin family. It is involved in various sterile inflammatory processes such as diabetic complications, chronic inflammatory diseases, atherosclerosis, and chronic neurodegenerative disease, such as Alzheimer’s disease [32]. A variety of tissues and cells express this PRR, including vascular endothelial, bronchial, and pulmonary cells; vascular smooth muscle cells; neurons; and many others [33,34]. Knowing the presence of DAMPs in the ocular surface of dry eye syndrome and the involvement of RAGE in many inflammatory diseases, it seemed interesting to us to study the implication of RAGE in the development (acute) or maintenance (chronic) of epithelial inflammation. In our work, the addition of the RAGE inhibitor (RAP) to the hyperosmolar medium of the in vitro model (HCE and WKD) did not show any decrease of upregulated cytokines. These data demonstrate that this receptor does not appear to be involved in ocular surface inflammation as an autocrine pathway. (This is also the case for the well-known TLR4 receptors described in Appendix A in this work.) Following these first results and considering that other receptors are able to bind these molecules, we globally test the autocrine actions of these alarmins released on epithelial cells. Using recombinant proteins on WKD and HCE cells, we demonstrated that neither of these alarmins separately induce inflammatory cytokines on ocular surface epithelia. Therefore, it would appear that the DAMPs studied here have no direct and autocrine inflammatory impact on the ocular surface epithelia.

Nevertheless, we pushed the study a little further by looking at the immune cells that can act on the ocular surface following acute inflammation. Indeed, we have highlighted the increased presence of MCP1 at surface epithelia in case of hyperosmolarity. MCP1 is known to be a chemokine which attracts particular monocytes and macrophages [35]. By secreting this pro-inflammatory chemokine, conjunctival and corneal cells could be responsible for the attraction of immune cells, namely, monocytes/macrophages that will further nourish the inflammatory process on the ocular surface and be a part of chronic inflammation. Indeed, macrophages have a primordial impact on inflammation, and they are among the first cells to reach the inflammatory site. Several actions are attributed to them, such as the ability to phagocytose cellular debris, pathogens, and apoptotic cells and the presentation of antigens to lymphocyte cells, linked to adaptative immunity. Macrophages are ready to rapidly produce large amounts of inflammatory cytokines in response to danger signals. This can induce a process of amplification and prolongation of the inflammation signal, which can lead to chronic inflammation [36].

Therefore, we were also interested in the impact of our previously characterized alarmins as potential attractive chemokines. Using THP1 cells, we demonstrated that not only MCP1 (internal positive control) but also S100A4, S100A9, and HMGB1 have the ability to attract macrophages. We then demonstrated that this action of alarmins on macrophages occur via the RAGE pathway. Globally, our data are in agreement with the literature, where a macrophagic attractant action has also been demonstrated for S100A8, one member of the S100 family [37,38]. It has already been shown that RAGE is important for the migration of dendritic cells and tumor cells [39,40,41,42]. Thus, these results highlight the potential importance of RAGE and some of its ligands in macrophage attraction in dry eye pathology as a paracrine manner following the activation of epithelial cells. To complete these results, it would be interesting to study the inflammatory activation of macrophages via the RAGE–ligands axis. Nevertheless, it has already been mentioned in the literature that these alarmins have the capacity to induce the expression of pro-inflammatory cytokines by macrophages [43,44,45,46]. This finding is in total agreement with the fact that in the dry eye syndrome, alarmins and MCP1 have the ability to attract and activate macrophages that are undoubtedly implied in the vicious circle maintenance. All these results obtained from cell lines should be compared and confirmed with primary human conjunctival cells, human corneal cells, and primary macrophages (i.e., peripheral blood mononuclear cells).

In summary, our results highlight the important role of corneal and conjunctival cells in the response of the ocular surface to a modification of the TF (i.e., dry eye hyperosmolarity). These cells are able to secrete specific danger molecules (alarmins) and pro-inflammatory mediators using NFAT5 transcription factor. Such action could be the start of an acute inflammatory reaction that will be transformed at term to an acute one following paracrine actions of MCP1 and alarmins on macrophages. Finally, the description of the RAGE receptor’s action and activation at the macrophage level looks like an interesting target for the treatment of this pathology.

## 4. Materials and Methods

### 4.1. Reagents

Cell culture media, cell transfection opti-MEM (1×) medium, epithelial growth factor (PHG0311), fetal bovine serum (FBS), insulin transferrin selenium, Pierce BCA Protein Assay Kit (23225), Lipofectamine 3000 Transfection Reagent (L3000008), and Lipofectamine RNAiMAX Transfection Reagent (13778150) were obtained from ThermoFisher Scientific (Illkirch-Graffenstaden, France). Antibiotics (streptomycin, penicillin, and amphotericin B) and glutamine were purchased from Eurobio Scientific (Les Ulis, France). Dimethyl sulfoxide (DMSO), HMGB1 (SRP6265, a mixture of different forms), and Fibronectin (F1141) were purchased from Sigma-Aldrich (Saint-Quentin-Fallavier, France). Human S100A4 (4137-S4), S100A8 (9876-S8), S100A9 (9254-S9), and MCP1 (279-MC-010/CF) recombinant proteins were obtained from R&D systems (Noyal-Châtillon-sur-Seiche, France). RAP (RAGE inhibitor, 553031) and TAK-242 (TLR4 inhibitor, 243984-11-4) were purchased from Merck (Saint-Quentin-Fallavier, France). LightCycler 480 SYBR Green I Master (04887352001) was provided by Roche (Meylan, France).

### 4.2. Cell Culture

Cells were obtained from ATCC (Manassas, VA, USA) and cultured under standard conditions (5% CO_2_, 95% humidified air, 37 °C). Human corneal epithelial cells (HCE) cells (CRL11135) were cultured, as previously described in Gross et al. [47], in DMEM supplemented with 10% heat-inactivated FBS, 2% glutamine, 5 µg/mL insulin transferine selenium, 0.1 µg/mL cholera toxin, 1% antibiotics (ampicillin 100 U/mL, streptomycin 100 µg/mL, and amphotericin B 25 µg/mL), 10 ng/mL epithelial growth factor, and 0.5% dimethyl sulfoxide. The Wong–Kilbourne derivative of Chang conjunctival cells (WKD) (CCL-20) were cultured in Dulbecco’s modified Eagle’s Medium culture supplemented with 1% antibiotics, 2% glutamine, and 10% heat-inactivated FBS. The THP-1 cell line (TIB-202) was cultured in a RPMI 1640 culture medium supplemented with 1% antibiotics, 2% glutamine, and 10% heat-inactivated FBS. The differentiation of THP1 into macrophages was carried out using a treatment with 6 ng/mL PMA (phorbol 12-myristate 13-acetate, ab120297, Abcam, Cambridge, UK).

#### 4.2.1. In Vitro Hyperosmolar Stress Experiment

Epithelial cells (WKD and HCE) were cultured in a six-well plate. When cells were at approximately 80–90% confluence, the culture medium was replaced with fresh medium +/− 70 mM or 90 mM NaCl to increase osmolarity for 6, 12, 24, or 48 h. In addition to hyperosmolarity, cells were treated with or without RAGE and TLR4 inhibitors (RAP and TAK, respectively) for 24 or 48 h. Cells were collected with trypsin (0.5%), centrifuged, and stored at −80 °C before use. The culture supernatant was also collected and stored at −80 °C before use. Osmolarity status was checked by osmometer measurements to validate the presence of hyperosmolarity.

#### 4.2.2. Alarmin Treatments

Epithelial cells (WKD and HCE) were cultured in a six-well plate. When cells were at approximately 80–90% confluence, the culture medium was replaced with fresh medium and treated with recombinant S100A4 (0.1 and 1 µg/mL), S100A8 (0.1 and 1 µg/mL), S100A9 (0.1 and 1 µg/mL), and/or HMGB1 (100 and 300 ng/mL). The culture supernatant and cells were collected and stored at −80 °C before use.

### 4.3. Quantitative RT-PCR

Total RNA was isolated from HCE and WKD cells with the NucleoSpin RNA Kit (Macherey–Nagel, Hoerdt, France) following the manufacturer’s instructions. Reverse transcription was carried out from 1 μg of RNA using a High-Capacity cDNA Reverse Transcription Kit (ThermoFisher Scientific, Illkirch-Graffenstaden, France) following the manufacturer’s instructions. All PCR primers were designed in the laboratory (Table 1). Quantitative RT-PCR using LightCycler^®^ 480 SYBR Green I Master (Roche, Meylan, France) was performed to evaluate the expression of these genes, using the housekeeping genes RSP17 (ribosomal protein S17) and RPL0 (36b4) as a control. Standard curves were used to quantify the amount of amplified transcript. Results were normalized to the geometric mean of the human housekeeping genes RPL0 (36b4) and RPS17 (acidic ribosomal phosphoprotein P0 and ribosomal protein S17, respectively) as recommended by the MIQE guidelines [48].

### 4.4. Western Blot

The extraction of total proteins for cells was carried out with RIPA, and concentrated protein supernatants were measured using a Pierce BCA protein assay kit. Proteins were separated via 4–15% Mini-PROTEAN^®^ TGX Stain-Free Precast Gel (Bio-Rad, Marnes-la-Coquette, France). The transfer was performed on a nitrocellulose membrane (Bio-Rad) and saturated for over 90 min with 5% milk in tris-buffered saline (TBS) 1X at room temperature. Then, the membrane was incubated overnight at 4 °C with the first antibody against NFAT5 (1:600, sc-398171, Santacruz) or HMGB1 (1:10,000, ab79823, Abcam) diluted in 5% milk in TBS 1X 0.1% tween-20. The membrane was washed with TBS 1X 0.1% tween-20 and then incubated for 90 min at RT with peroxidase-conjugated secondary antibody anti-mouse or anti-rabbit, respectively (1/10,000). After washing, the revelation was completed using an ECL clarity kit (Clarity Western ECL Substrate, BIO-RAD, Marnes-la-Coquette, France) for Western blot and integrated on the ChemiDoc imaging system (Bio-Rad). The relative intensities of the protein bands were analyzed using Image Lab 6.1 software (BIO-RAD, Marnes-la-Coquette, France). Total protein normalization (Bio-Rad), a method allowing normalization using the total protein loaded, was used to normalize the Western blot results.

### 4.5. Supernatant Protein Concentration

Before the Western blot assays, ELISA, and treatment with a conditioned medium, supernatants were concentrated into 2 kDa centrifugal filter units (Vivacon^®^ 500, Sartorius, Aubagne, France) for protein concentration and purification following the manufacturer’s instructions.

### 4.6. Cytokine Multiplex Assay

The release of IL6, IL8, and MCP1 in the culture media was tested using a Human Luminex^®^ Discovery Assay (Biotechne, Rennes, France) based on Luminex^®^ xMAP^®^ technology according to the manufacturer’s instructions (using 1:2 diluted media). The release of TNFα in the culture media was tested using automated multiplex immunoassays in ELLA^TM^ (San Jose, CA, USA) with 1:2 diluted media. Finally, the ratio “treated/non-treated” was reported.

### 4.7. ELISA

Human S100A4 protein levels were quantified by ELISA kits (ab283547, Abcam, Cambridge, UK), human S100A8 and S100A9 protein levels were quantified by a DuoSet ELISA kit (respectively DY4570-05 and DY5578, R&D systems), and human S100A8/A9 (calprotectin) protein levels were quantified using ELISA kits (EH62RB, Invitrogen, Waltham, MA, USA) following the manufacturers’ instructions. Protein levels were assessed in media from hyperosmolar experiments HCE and WKD. Data were collected using the Multiskan^®^ spectrophotometer Spectrum. For S100A4 ELISA kits, WKD culture supernatants were used diluted to 1:10 and HCE used undiluted. For S100A8/A9 ELISA kits, WKD culture supernatants were used concentrated and HCE used 1:2 diluted. For S100A8 DuoSet ELISA kits and S100A9 DuoSet ELISA kits, culture supernatants (WKD and HCE) were used undiluted.

### 4.8. Small Interfering RNA Transfection of Epithelial Cells (HCE and WKD)

For hyperosmolar stress experiments, NFAT5 expression was decreased in HCE and WKD cells using small interfering RNA (siRNA). Epithelial cells at 80% confluence in six-well plates were transfected for 24 h with 25 nM commercial siRNA against NFAT5 (M-009618-01; Dharmacon, Lafayette, LA, USA) or with 25 nM of non-targeting siRNA (siRNA control; D-001206-14; Dharmacon) in 150 µL opti-MEM (1×) medium mixed with 9 µL Lipofectamine RNAiMAX Reagent in 150 µL opti-MEM (1×) medium. The decrease in NFAT5 upon treatment with siRNA was monitored via RT-qPCR.

### 4.9. NFAT5 Gene Reporter Luciferase Assay

For hyperosmolar stress experiments, cells were transfected with two plasmids using Lipofectamine 3000. The first vector was NFAT5 luciferase (Plasmid #14110; Addgen, Watertown, MA, USA), which contains the luciferase reporter gene where transcription depends on ORE fixation sites. The second vector was PCH110 (vector kindly gifted by the laboratory of Prof. Pierre CHAMBON [IGBMC, Illkirch-Grafenstaden, France]) containing the gene LAC-Z and encoding β-galactosidase used as an internal transfection control. The amount of each vector was 1 µg of NFAT5 luciferase and 0.15 µg of PCH110. After 4 h of transfection in an OPTIMEM medium, the transfection media was replaced by a medium containing the treatments (see treatments paragraph 4.2.1). After 24 h of incubation at 37 °C under 5% CO_2_, cells were collected and kept at −20 °C.

Measurement of luciferase activity: The cells were lysed in 120 µL of 1X lysis buffer from the Luciferase Reporter Gene Assay kit (11814036001, Roche, Meylan, France). After 5 min of incubation at room temperature, the cellular debris were eliminated by centrifugation (3 min, 13,000 rpm), and supernatant (20 µL) was mixed with 50 µL buffer containing the enzyme substrate. Under the action of luciferase, the substrate was transformed into a luminescent product for which the measured intensity—using a FB12 luminometer (Berthold, Thoiry, France)—was proportional to the activity of the luciferase, which itself depends on NFAT5 activity.

Measurement of β-galactosidase activity: In a 96-well plate, 50 µL of supernatant was incubated for 10 min at 37 °C in the dark with 50 µL of the reagent from the Mammalian, β-galactosidase Assay kit (75707, Thermo Scientific, Waltham, MA, USA) containing the enzyme substrate. Βeta-galactosidase turns this substrate a yellow color. Its optical density, measured at 405 nm using a Multiskan^®^ spectrophotometer Spectrum (Thermo Scientific, Waltham, MA, USA), is proportional to the cells transfected. Results were then obtained by carrying out the Luciferase/β-galactosidase report. The final results are expressed as the fold change between the treatment and control group (NT).

### 4.10. Immunofluorescence

After permeabilization in PBS 1X/FBS 10%/Triton 0.1% for over 90 min, the primary antibody against NFAT5 (1:400, PA1-023, Thermo Fisher, Waltham, MA, USA) was applied to the cells overnight at 4 °C. After three washes in the permeabilization buffer, the secondary antibody anti-rabbit Alexa Fluor 488 (1:400, A21206, Life Technologies, Carlsbad, CA, USA) was incubated for 2 h at room temperature. Slides were washed three times in TWEEN^®^ PBS 1X and incubated with Hoechst (15 min, dilution in PBS 1X 1/10,000; bisBenzimide H, 33258, Sigma-Aldrich, St. Louis, MO, USA). Finally, slides were mounted with CitiFluor^TM^ Tris-MWL 4–88 (Electron Microscopy Science, Hatfield, PA, USA) and examined under an LSM800 Confocal microscope (magnification ×200). For negative controls, incubation without the primary antibody was performed.

### 4.11. Chemotaxis Aassay

The chemotaxis assay of macrophages was assessed with CytoSlect^TM^ 24-well cell migration assay (CBA-101; 8 µm fluorometric format; CliniSciences, Nanterre, France). THP1 cells were differentiated on the top of the insert and coated with fibronectin at 5 µg/mL in normal medium with PMA for 48 h. After 48 h under the insert, 500 µL of the chemoattractant media were added, and on the top (macrophages presence), fresh medium was added. The chemotractants used were S100A4 at 2 ng/mL, S1008 at 3 ng/mL, S100A9 at 2 ng/mL, HMGB1 at 100 ng/mL, and MCP1 at 100 ng/mL. In the case of the inhibitors, they were added to the top and bottom of the insert at 7 µg/mL for TAK and 12.7 µg/mL for RAP. After 48 h incubation, cells that had passed through the membrane (8 µm pore size) were dislodged with a detachment solution. Dislodged cells were stained with CyQuant GR Dye diluted in a lysis buffer and quantified by fluorescence measurement at 480–520 nm. Cells at the top of the membrane were also dislodged and stained for the normalization.

### 4.12. Statistical Analysis

The data were expressed as the mean ± standard error of the mean and were an average of the duplicates of at least three or four independent experiments. Given that the results did not follow a normal distribution, a comparison of means was performed using nonparametric tests. First, a one-way ANOVA Kruskal–Wallis was performed to study a global comparison between all groups, followed by multiple comparisons with Dunn’s correction for more than two groups using PRISM 8 (GraphPad Software Inc., San Diego, CA, USA). For all studies, the values were considered significantly different at *p* < 0.05 (*), *p* < 0.01 (**), and *p* < 0.001 (***).

## Figures and Tables

**Figure 1 ijms-24-11052-f001:**
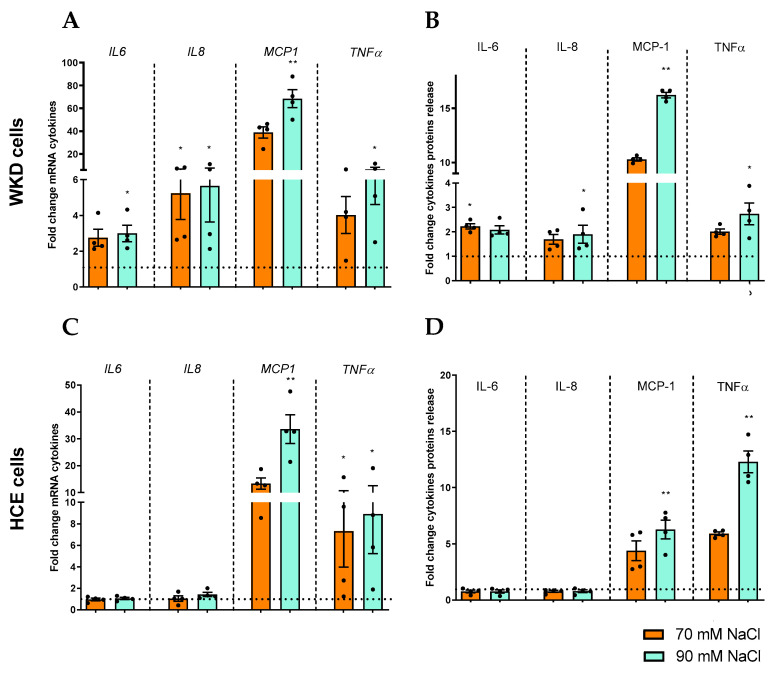
Induction of cytokines by hyperosmolarity in different ocular surface epithelial cells. Cells were exposed to a hyperosmolar medium (70 mM NaCl [orange] or 90 mM NaCl [green]) for 24 h, and (**A**) mRNA expressions of IL6, IL8, TNFα, and MCP1 were analyzed via RT-qPCR, and (**B**) after 48 h of hyperosmolarity exposition, cytokine production and release were analyzed viaMultiplex in a Wong–Kilbourne derivative of the Chang conjunctival (WKD) cell line. (**C**) The mRNA expression of cytokines were analyzed (**D**) and protein release in a human corneal epithelial (HCE) cell line. The experiment was repeated four times (n = 4). Results are expressed as a fold change between treated and non-treated (NT) conditions and were analyzed via ANOVA statistical test (Kruskal–Wallis) followed by Dunn’s post hoc test: * *p* = 0.05 and ** *p* = 0.01. The horizontal line corresponds to the value of the non-treated cells reported as 1.

**Figure 2 ijms-24-11052-f002:**
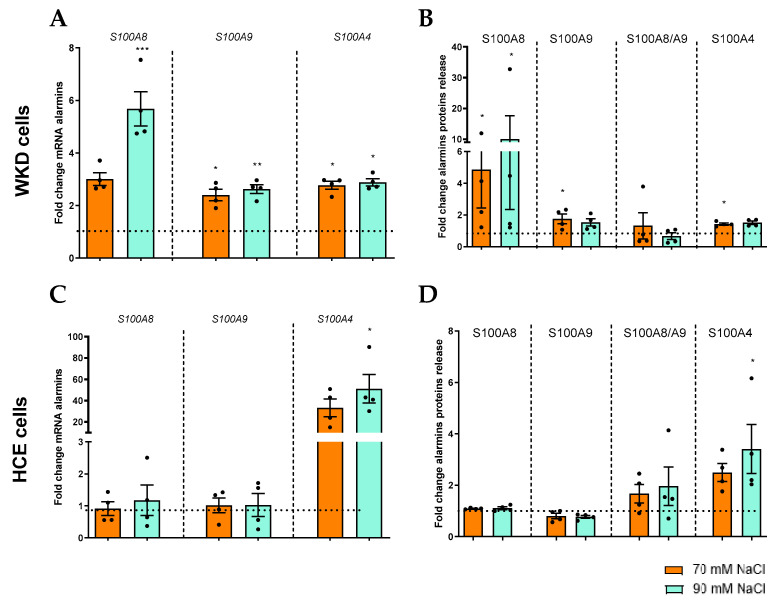
Induction of S100 proteins by hyperosmolarity in different ocular surface epithelial cells. Cells were exposed to a hyperosmolar medium (70 mM NaCl [orange] or 90 mM NaCl [green]) for 24 h and (**A**) analyzed for mRNA expressions of S100A4, S100A8, and S100A9 via RT-qPCR. (**B**) After 48 h of hyperosmolarity exposition, the protein releases of S100A4, S100A8, S100A9 and S100A8/A9 were analyzed via ELISA in a Wong–Kilbourne derivative of the Chang conjunctival (WKD) cell line. (**C**) The mRNA expression of alarmins were analyzed (**D**) and protein release in a human corneal epithelial (HCE) cell line. The experiment was repeated four times (n = 4). Results are expressed as a fold change between treated and non-treated (NT) conditions and were analyzed via ANOVA statistical test (Kruskal–Wallis), followed by Dunn’s post hoc test: * *p* = 0.05, ** *p* = 0.01 and *** *p* = 0.001. The horizontal line corresponds to the value of the non-treated cells reported as 1 (n = 4).

**Figure 3 ijms-24-11052-f003:**
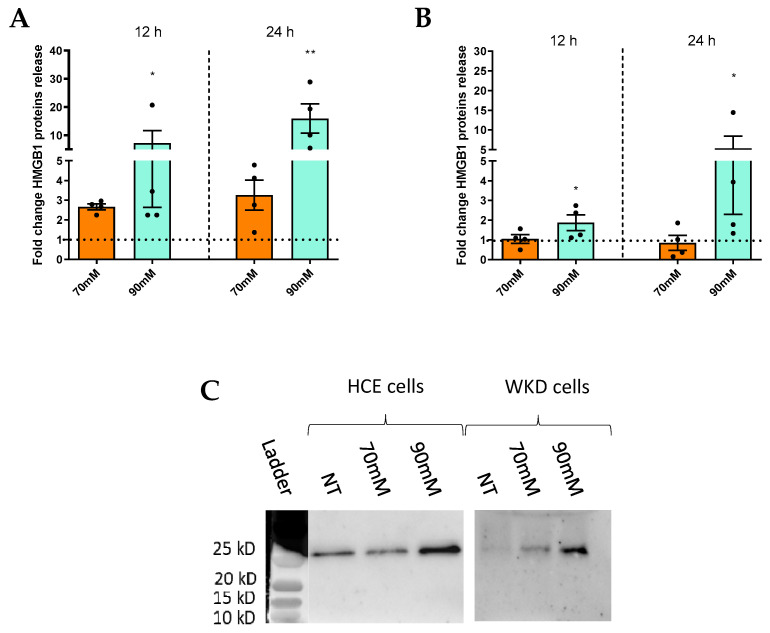
Induction of HGMB1 release by hyperosmolarity in different ocular surface epithelial cells. Cells were exposed to a hyperosmolar medium (70 mM NaCl [orange] or 90 mM NaCl [green]) for 12 and 24 h, and the medium was used for WB analyses of HMGB1 release for (**A**) a Wong–Kilbourne derivative of the Chang conjunctival (WKD) cell line and (**B**) a human corneal epithelial (HCE) cell line. (**C**) Western blot results after 24 h of hyperosmolarity for HCE and WKD cell line. Experiments were repeated four times (n = 4). Results are expressed as a fold change between treated and non-treated (NT) conditions and were analyzed via ANOVA statistical test (Kruskal–Wallis), followed by Dunn’s post hoc test: * *p =* 0.05 and ** *p* = 0.01. The horizontal line corresponds to the value of the non-treated cells reported as 1 (n = 4).

**Figure 4 ijms-24-11052-f004:**
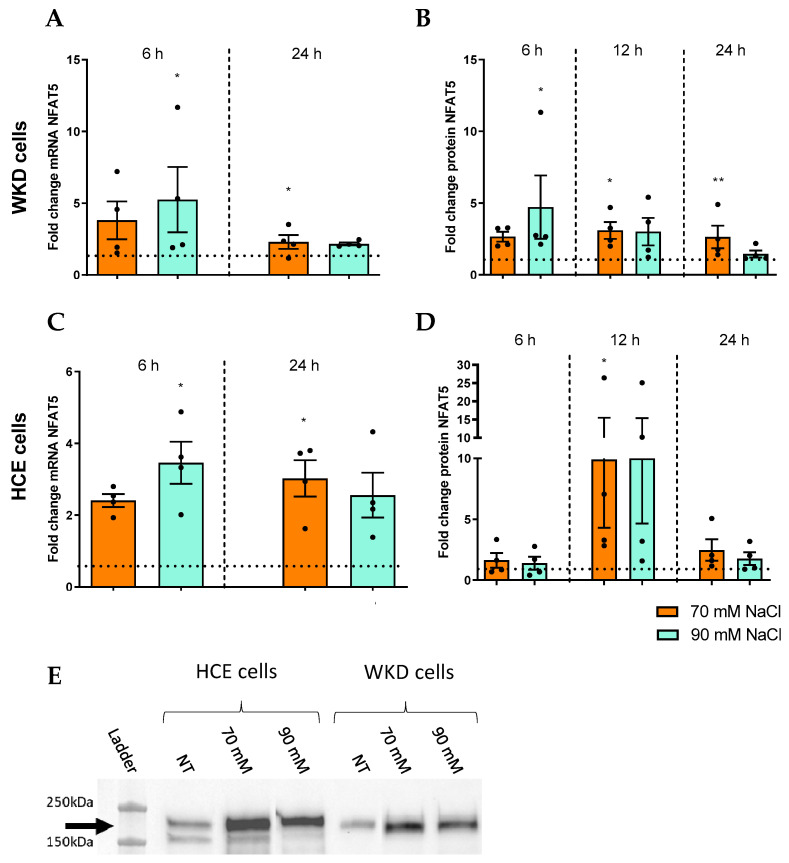
Induction of NFAT5 by hyperosmolarity in different ocular surface epithelial cells. Cells were exposed to a hyperosmolar medium (70 mM NaCl [orange] or 90 mM NaCl [green]) for 6, 12, and 24 h and (**A**) analyzed for NFAT5 mRNA expression via RT-qPCR and (**B**) for protein expression by Western blot in a Wong–Kilbourne derivative of the Chang conjunctival (WKD) cell line. (**C**) The mRNA expression of NFAT5 and (**D**) protein expression were analyzed in a human corneal epithelial (HCE) cell line. (**E**) Western blot results after 12 h of hyperosmolarity for HCE and WKD cell line. The experiment was repeated four times (n = 4). Results are expressed as a fold change between treated and non-treated (NT) conditions and were analyzed via ANOVA statistical test (Kruskal–Wallis), followed by Dunn’s post hoc test: * *p* = 0.05 and ** *p* = 0.01. The horizontal line corresponds to the value of the non-treated cells reported as 1.

**Figure 5 ijms-24-11052-f005:**
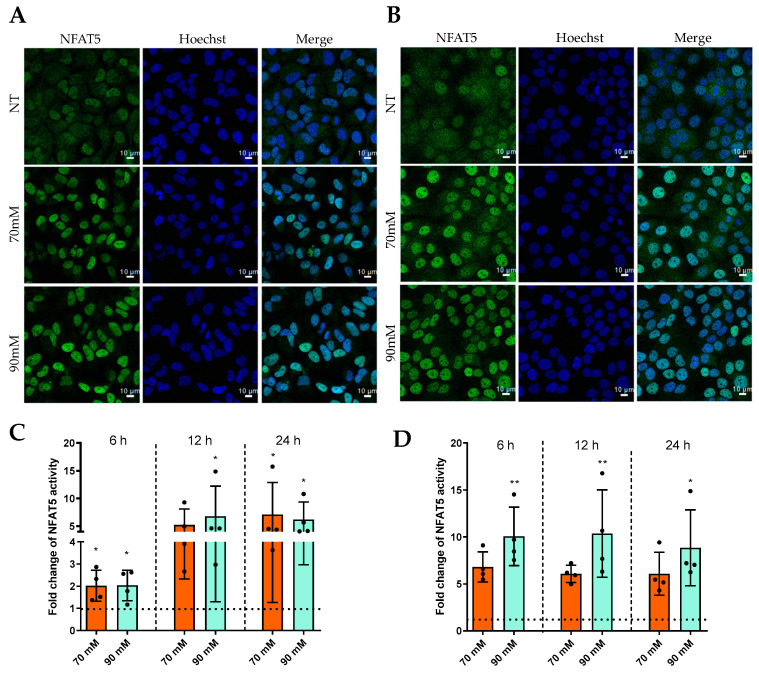
Activation of NFAT5 by hyperosmolarity in different ocular surface epithelial cells. Cells were exposed to a hyperosmolar condition for 6 h, and NFAT5 localization was analyzed via immunofluorescence in (**A**) a Wong–Kilbourne derivative of the Chang conjunctival (WKD) cell line and (**B**) a human corneal epithelial (HCE) cell line. After 6, 12, and 24 h of hyperosmolarity exposition, NFAT5 transcriptional activity was assessed via a bioluminescence reaction using the Luciferase Reporter Gene Assay kit (Roche Applied Science) in (**C**) the WKD cell line and (**D**) the HCE cell line. The experiment was repeated four times (n = 4). Results are expressed as a fold change between treated and non-treated (NT) conditions and were analyzed via ANOVA statistical test (Kruskal–Wallis), followed by Dunn’s post hoc test: * *p* = 0.05 and ** *p* = 0.01. The horizontal line corresponds to the value of the non-treated cells reported as 1.

**Figure 6 ijms-24-11052-f006:**
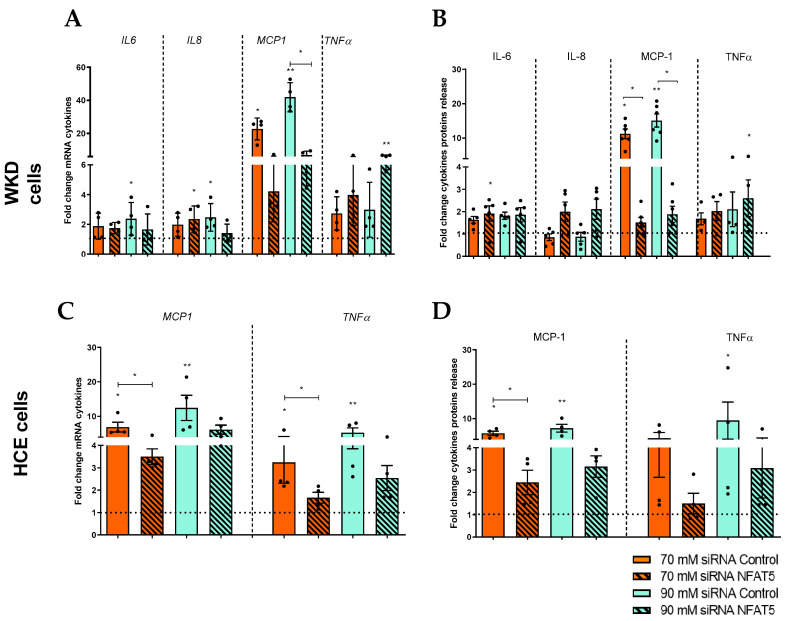
Implication of NFAT5 in ocular surface epithelial cell inflammation. After transfection with siRNA against NFAT5 (hatched boxes) or siRNA control, cells were exposed to a hyperosmolar medium (70 mM NaCl [orange] or 90 mM NaCl [green]) for 24 h and (**A**) then analyzed for mRNA expression of cytokines via RT-qPCR and (**B**) exposed for 48h to hyperosmolarity for cytokine protein release analysis in a Wong–Kilbourne derivative of the Chang conjunctival (WKD) cell line. (**C**) The mRNA expression of cytokines and (**D**) protein release were analyzed in a human corneal epithelial (HCE) cell line. The experiment was repeated four times (n = 4). Results are expressed as a fold change between treated and non-treated (NT siRNA control) conditions and were analyzed via ANOVA statistical test (Kruskal–Wallis), followed by Dunn’s post hoc test: * *p* = 0.05 and ** *p* = 0.01. The horizontal line corresponds to the value of the non-treated cells reported as 1.

**Figure 7 ijms-24-11052-f007:**
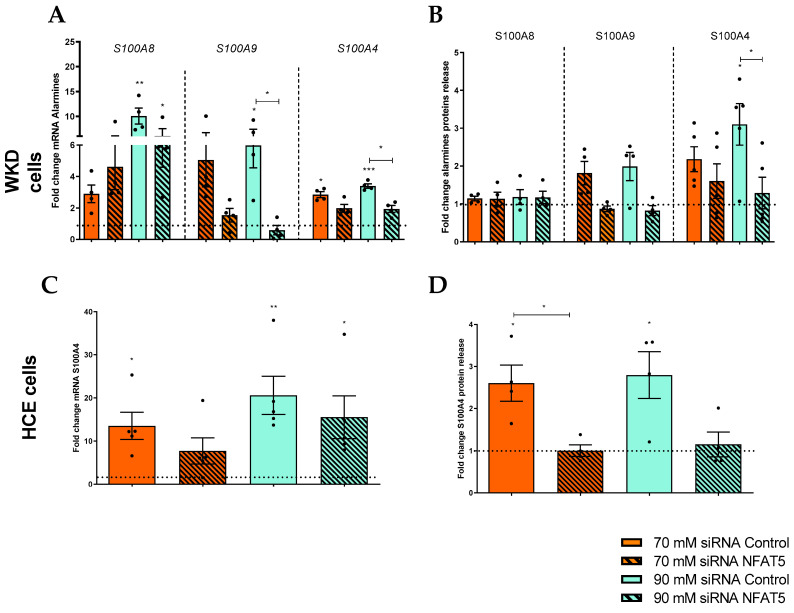
Implication of NFAT5 in ocular surface epithelial cell alarmins. After transfection with siRNA control or NFAT5 (hatched boxes), cells were exposed to a hyperosmolar medium (70 mM NaCl [orange] or 90 mM NaCl [green]) for 24 h and (**A**) then analyzed for mRNA expression of alarmins via RT-qPCR and (**B**) exposed for 48h to hyperosmolarity for alarmin protein release analysis in a Wong–Kilbourne derivative of the Chang conjunctival (WKD) cell line. (**C**) The mRNA expression of cytokines and (**D**) protein release were analyzed in a human corneal epithelial (HCE) cell line. The experiment was repeated four times (n = 4). Results are expressed as a fold change between treated and non-treated (NT siRNA control) conditions and were analyzed via ANOVA statistical test (Kruskal–Wallis) followed by Dunn’s post hoc test: * *p* = 0.05, ** *p =* 0.01 and *** *p* = 0.001. The horizontal line corresponds to the value of the non-treated cells reported as 1.

**Figure 8 ijms-24-11052-f008:**
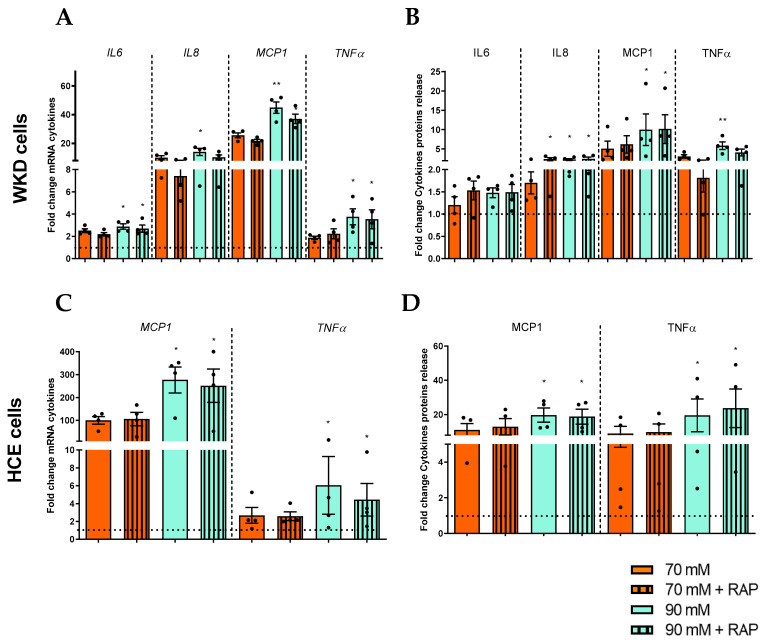
Implication of RAGE on the cytokines release of epithelial cells. Cells were exposed to a hyperosmolar medium (70 mM NaCl [orange] or 90 mM NaCl [green]), with or without RAP (RAGE inhibitor [hatched boxes]), for 24 h and (**A**) then analyzed for mRNA expression of cytokines via RT-qPCR and (**B**) exposed for 48 h to hyperosmolarity for cytokine protein release analysis in a Wong–Kilbourne derivative of the Chang conjunctival (WKD) cell line. (**C**) The mRNA expression of cytokines and (**D**) protein release were analyzed in a human corneal epithelial (HCE) cell line. The experiment was repeated four times (n = 4). Results are expressed as a fold change between treated and non-treated (NT) conditions and were analyzed via ANOVA statistical test (Kruskal–Wallis), followed by Dunn’s post hoc test: * *p* = 0.05 and ** *p* = 0.01. The horizontal line corresponds to the value of the non-treated cells reported as 1.

**Figure 9 ijms-24-11052-f009:**
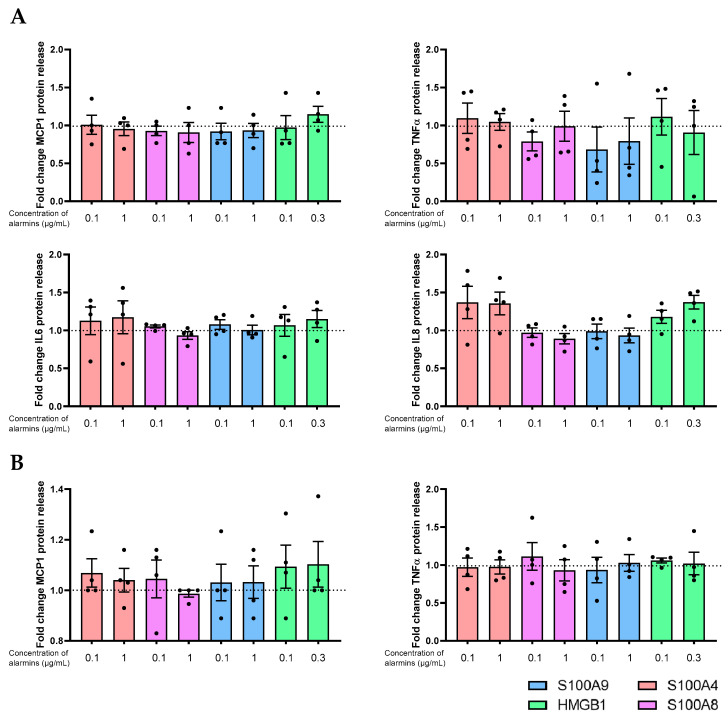
Action of DAMPs on the epithelial cell release of cytokines. Epithelial cells were exposed to a normal medium with protein recombinant (S100A4 [pink], S100A8 [purple], or S1009 [blue] at 0.1 and 1 µg/mL or HMGB1 [green] at 100 or 300 ng/mL) for 48 h and then analyzed for protein release of IL6, IL8, TNFα, and MCP1 via ELLA and Multiplex in (**A**) a Wong–Kilbourne derivative of the Chang conjunctival cell (WKD) line and (**B**) a human corneal epithelial (HCE) cell line. The experiment was repeated four times (n = 4). Results are expressed as a fold change between treated and non-treated (NT) conditions and were analyzed via ANOVA statistical test (Kruskal–Wallis), followed by Dunn’s post hoc test. The horizontal line corresponds to the value of the non-treated cells reported as 1.

**Figure 10 ijms-24-11052-f010:**
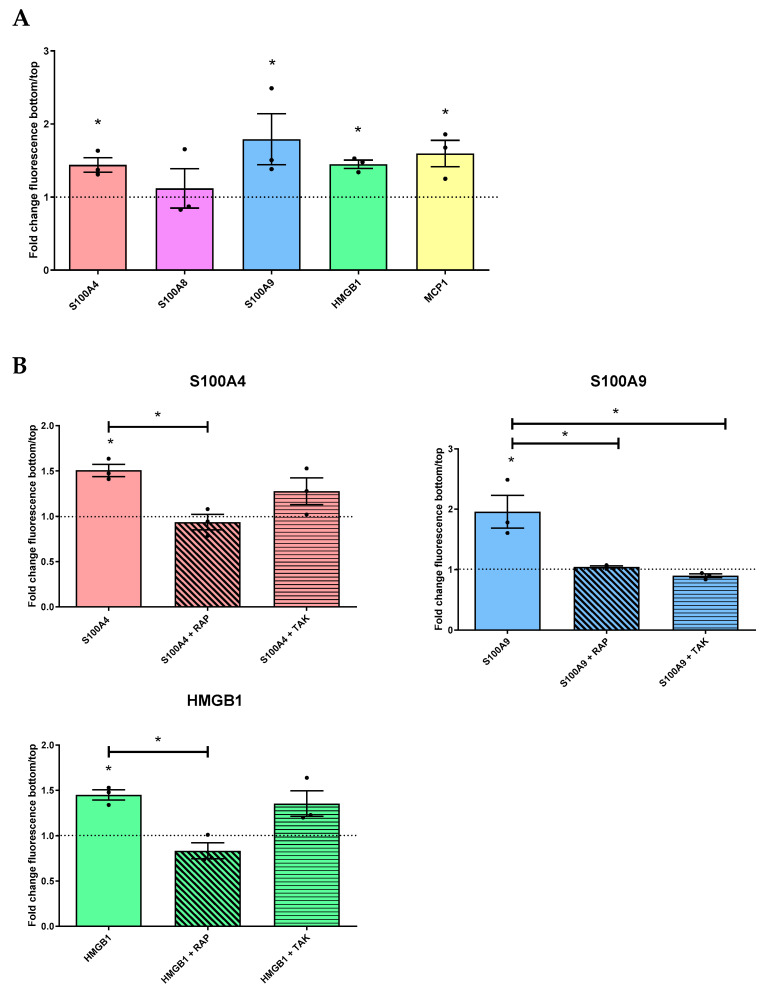
Effects of DAMPs and MCP1 in macrophage migration. In the CytoSlect^TM^ 24-well cell migration assay, THP1 was differentiated for 48 h. (**A**) Next, macrophages were exposed to S100A4 (2 ng/mL [pink]), S100A8 (3 ng/mL [purple]), S100A9 (2 ng/mL [blue]), HMGB1 (100 ng/mL [green]), or MCP1 (100 ng/mL [yellow]). (**B**) Macrophages were exposed to the same DAMPs with or without RAGE inhibitors (RAP [angled hatches]) and TLR4 (TAK [horizontal hatches]). The experiment was repeated three times (n = 3). The quantification of cells in the top and bottom of the insert was measured via fluorescence. Results are expressed as a fold change between treated and non-treated (NT) conditions and were analyzed via ANOVA statistical test (Kruskal–Wallis), followed by Dunn’s post hoc test. * *p* = 0.05. The horizontal line corresponds to the value of the non-treated cells reported as 1.

**Table 1 ijms-24-11052-t001:** Forward and reverse primer sequences used for RT-PCR amplification of human genes.

Gene	Sequence 5′-3′ (F: Forward; R: Reverse)	Product Size (bp)	Hybridization Temperature (°C)
hsRPS17	F: TGCGAGGAGATCGCCATTATC	169	61
	R: AAGGCTGAGACCTCAGGAAC		
hsRPL0	F: AGGCTTTAGGTATCACCACT	219	61
	R: GCAGAGTTTCCTCTGTGATA		
hsNFAT5	F: ACAGTAAAGCTGGAAGGCCA	185	61
	R: TTGCTAGGATCAAGGCCGAC		
hsS100A9	F: ACACTCTGTGTGGCTCCTCG	166	61
	R: CGCACCAGCTCTTTGAATTCC		
hsS100A4	F: GGACAGCAACAGGGACAACGA	101	61
	R: TATCTGGGAAGCCTTCAAAG		
hsS100A8	F: TAAAGGGGAATTTCCATGCCGT	137	61
	R: GTTAACTGCACCATCAGTGTTG		
hsIL8	F: TGATTTCTGCAGCTCTGTGTG	154	61
	R: TCTGTGTTGGCGCAGTGTGG		
hsIL6	F: AATGAGGAGACTTGCCTGGTG	143	61
	R: AGGAACTGGATCAGGACTTTTG		
hsMCP1	F: ATAGCAGCCACCTTCATTCCC	185	61
	R: ATCTCCTTGGCCACAATGGTC		
hsTNFα	F: AGGGACCTCTCTCTAATCAGC	168	61
	R: TCTCAGCTCCACGCCATTGG		

## Data Availability

Not applicable.

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
