# Peer review of "Inflammation of Dry Eye Syndrome: A Cellular Study of the Epithelial and Macrophagic Involvement of NFAT5 and RAGE"

_ijms, 2023, doi:10.3390/ijms241311052_

Round 1

Reviewer 1 Report

In the current study, authors have investigated the role of two inflammatory factors NFAT5 and RAGE by using two in vitro cell lines, human corneal epithelial and Wong Kilbourne derivative cells as dry eye disease model. Overall study is well designed and presented evidence justifies the conclusions made. The reviewer has following concerns.

Grammatical errors are present throughout the manuscript. These errors drastically change, in some cases, the meaning of sentences. For example, lines 144-146: “This mRNA overexpression....of the increase of NFAT5 protein level”, here of should be for.

Results are heavily focused on quantitative assays like qPCR and luciferase assays. Some fold changes (~1-2 fold) are statistically significant with minimal difference compared to that of NT while others with a higher fold difference (~3-6 fold) are not significant. This suggests a high standard deviation between experiments and authors should present the results as bar graphs with dot plots to give more insights and easy reference instead of box plots. Similar is the case for luciferase assays in Fig 4C-D. How many times each experiments were replicated? Western blot pictures should be embedded with their loading control instead of in the supplementary files.

Manuscript needs extensive editing to correct grammatical errors.

Author Response

Reviewer #1 : Point by point answer from the authors

*Point 1: In the current study, authors have investigated the role of two inflammatory factors NFAT5 and RAGE by using two in vitro cell lines, human corneal epithelial and Wong Kilbourne derivative cells as dry eye disease model. Overall study is well designed and presented evidence justifies the conclusions made. The reviewer has following concerns.

Answer 1: The authors thank the reviewer for their kind comments towards the study and positive remarks concerning the manuscript.

*P2:  Grammatical errors are present throughout the manuscript. These errors drastically change, in some cases, the meaning of sentences. For example, lines 144-146: “This mRNA overexpression....of the increase of NFAT5 protein level”, here of should be for.

A2: First, the authors change “of” for “for” in the text as proposed by the reviewer (line 156). In fact, it was not the correct word. For the “Grammatical errors present throughout the manuscript”, the authors are somewhat surprise. In fact, as “non-native” English speakers, our research team used all the times since many years a Scientific English proofreading company called “Scribendi” (scribendi.com) before any scientific article submission. This one is always indicated in our manuscript (as also for this manuscript) and we do not encounter till now any English language remarks by other reviewers following the use of this service. Please find here one recent example of publication edited by such service (Physiological TLR4 regulation in human fetal membranes as an explicative mechanism of a pathological preterm case. Belville C, Ponelle-Chachuat F, Rouzaire M, Gross C, Pereira B, Gallot D, Sapin V, Blanchon L. Elife. 2022 Feb 4;11:e71521. doi: 10.7554/eLife.71521 /).  The authors mentioned the proofreading by Scribendi in the “Acknowledgments part” of the revised version (lines 668-669).

*P3: Results are heavily focused on quantitative assays like qPCR and luciferase assays. Some fold changes (~1-2 fold) are statistically significant with minimal difference compared to that of NT while others with a higher fold difference (~3-6 fold) are not significant. This suggests a high standard deviation between experiments and authors should present the results as bar graphs with dot plots to give more insights and easy reference instead of box plots. Similar is the case for luciferase assays in Fig 4C-D.

A3: We agree with the reviewer and the revised version of the manuscript include all revised figures as bar graph with dot plots.

*P4: How many times each experiments were replicated?

A4: Again, and according to the reviewer comments, we have revised the manuscript to include under each figures the number of replicates for each experiment. This modification was also made for the supplementary files. Moreover, the number of duplicates is mentioned in materials and methods "statistical analysis" part (lines 643-644).

*P5: Western blot pictures should be embedded with their loading control instead of in the supplementary files.

A5: As suggested by the reviewer, western blot pictures were included in Figure 3 and Figure 4. According to materials and method sections, our western-blot were analyzed using total proteins loading, that why we decided to only include in the revised version the specific protein bands (NFAT5 and HMGB1). The legends of the figures have been modified accordingly (lines 143-144 and 165-166).

Reviewer 2 Report

The manuscript by Henrioux et al. represents an in vitro study of the epithelial and macrophagic involvement of NFAT5 and RAGE. Although the study is mostly descriptive, it contains some novel data. Nonetheless, the study is well designed, the conclusions are supported by the data presented, while the manuscript is well written and easy to follow. Nonetheless I have some minor suggestions, which I hope will improve the presentation.

To simplify presentation easier for understanding of further reader, I would like to suggest the authors incorporating the names for cell lines within the figure 1. Moreover, logically panels A/C and B/D could be presented as combined panels for each cell line uniting mRNA and protein data.

For Figures 2, 4, 5, 6, 7: similar as above.

Fig. 7: the legend describing each sample is applicable to all the panels and, therefore, should be placed at the lower right corner of the hole figure.

Fig. 8, 9: similar as above.

Fig. 10: what different colors/hatching mean?

I did not find the sequences for siRNA used.

Author Response

Reviewer #2 : Point by point answer from the authors

*Point 1: The manuscript by Henrioux et al. represents an in vitro study of the epithelial and macrophagic involvement of NFAT5 and RAGE. Although the study is mostly descriptive, it contains some novel data. Nonetheless, the study is well designed, the conclusions are supported by the data presented, while the manuscript is well written and easy to follow. Nonetheless I have some minor suggestions, which I hope will improve the presentation.

Answer 1: The authors thank the reviewer for the positive comments towards the study and remarks concerning the manuscript.

*P2:  To simplify presentation easier for understanding of further reader, I would like to suggest the authors incorporating the names for cell lines within the figure 1.

A2: We agree with the reviewer and we have revised the manuscript to include the names of cell lines in Figure 1 (page 3), 2 (page 4), 4 (page 6), 6 (page 8), 7 (page 9) and 8 (page 11).

* P3:  Moreover, logically panels A/C and B/D could be presented as combined panels for each cell line uniting mRNA and protein data.

A3: Again, we follow the suggestion of the reviewer. We have revised the manuscript to include mRNA and protein data combined for each cell line in Figure 1 (page 3), 2 (page 4), 4 (page 6), 6 (page 8), 7 (page 9) and 8 (page 11).The legends of the figures have been modified accordingly.

* P4: Fig. 7: the legend describing each sample is applicable to all the panels and, therefore, should be placed at the lower right corner of the hole figure.

A4: We have revised the manuscript to include the legend at the lower right corner of all figures.

* P5: Fig. 10: what different colors/hatching mean?

A5: We have revised the manuscript to clarify the figure legend (lines 317 to 320).

* P6: I did not find the sequences for siRNA used.

A6 : The siRNAs used are marketed by Dharmacon. You can find the references in the materials and methods section "4.8 Small interfering RNA transfection of epithelial cells (HCE and WKD)" from line 577 to 584.Unfortunately, the private company does not publish the sequence used for its siRNA against NFAT5 on the website. Please find below the two references used as indicated in the material and methods section:

  • siRNA NFAT5 : M-009618-01; Dharmacon, Lafayette, LA, USA  (mixture of 4 siRNA).
  • - siRNA control : D-001206-14; Dharmacon, Lafayette, LA, USA.

Round 2

Reviewer 1 Report

Authors have addressed the comments satisfactorily.

Extensive editing of the english language in the manuscript is needed.